# Look Around and Find Out: OOD Detection with Relative Angles

## Abstract

Deep learning systems deployed in real-world applications often encounter data that is different from their in-distribution (ID). A reliable system should ideally abstain from making decisions in this out-of-distribution (OOD) setting. Existing state-of-the-art methods primarily focus on feature distances, such as k-th nearest neighbors and distances to decision boundaries, either overlooking or ineffectively using in-distribution statistics. In this work, we propose a novel angle-based metric for OOD detection that is computed relative to the in-distribution structure. We demonstrate that the angles between feature representations and decision boundaries, viewed from the mean of in-distribution features, serve as an effective discriminative factor between ID and OOD data. Our method achieves state-of-the-art performance on CIFAR-10 and ImageNet benchmarks, reducing FPR95 by $0.88\%$ and $7.74\%$ respectively. Our score function is compatible with existing feature space regularization techniques, enhancing performance. Additionally, its scale-invariance property enables creating an ensemble of models for OOD detection via simple score summation.

## 1 Introduction

A trustworthy deep learning system should not only produce accurate predictions, but also recognize when it is processing an unknown sample. The ability to identify when a sample deviates from the expected distribution, and potentially rejecting it, plays a crucial role especially in safety-critical applications, such as medical diagnosis (Fernando et al., 2021), driverless cars (Bogdoll et al., 2022) and surveillance systems (Diehl & Hampshire, 2002). The out-of-distribution (OOD) detection problem addresses the challenge of distinguishing between in-distribution (ID) and OOD data – essentially, drawing a line between what the system knows and what it does not.

Various approaches have been proposed for OOD detection, mainly falling into two categories: (i) methods that suggest model regularization during training (Lee et al., 2018a; Hendrycks et al.; Meinke & Hein), and (ii) post-hoc methods, which leverage a pre-trained model to determine if a sample is OOD by designing appropriate *score functions* (Peng et al., 2024; Hendrycks & Gimpel, 2022; Sun et al., 2022). Post-hoc methods are more advantageous for their efficiency and flexibility, as they can be applied to arbitrary pre-trained models without retraining. These approaches are often categorized based on the domain of their score functions, i.e., at which representational abstraction level they assess if a sample is OOD or not. Earlier techniques focus on measuring the model confidence in the logits space (Hendrycks & Gimpel, 2022; Liu et al., 2020), while the recent works employ distance-based scores (Sun et al., 2022; Sehwag et al., 2021) defined in the model feature space. While logit-based methods suffer from the overconfident predictions of neural networks (Minderer et al., 2021; Lakshminarayanan et al., 2017; Guo et al., 2017), the recent success of distance-based techniques highlights that the relationships in the latent space can provide a richer analysis.

A natural approach to feature representations is by checking their proximity to the decision boundaries (Liu & Qin, 2024). Conceptually, this can be related to identifying hard-to-learn examples in data-efficient learning (Joshi et al., 2024; Chen et al., 2023). OOD samples can be viewed as hard-to-learn since they do not share the same label distribution as ID data. The success of this approach has been directly showed in fDBD score from Liu & Qin (2024). However, our derivations revealed that the regularization term they use to incorporate ID statistics introduces an additional term that does not correlate with ID/OOD separation, ultimately hindering their performance.

In this work, we present Look Around and Find Out (LAFO), a novel approach that exploits the relationship between feature representations and classifier decision boundaries, in the context of the mean statistics of ID features. Unlike the earlier techniques, LAFO introduces a new angle-based measure that calculates the angles between the feature representations and their projection onto the decision boundaries, relative to the the mean statistics of ID features. Changing reference frame to the mean of ID features adds another layer of discriminatory information to the score, as it naturally incorporates the ID statistics to the distance notion, exploiting the disparity between ID and OOD statistics. Moreover, the scale-invariant nature of angle-based representations, as similarly observed in (Moschella et al.), allows us to aggregate the confidence scores from multiple pre-trained models simply by summing their LAFO scores. This enables to have a score that can be single model based or extended to ensemble of models. In summary, our key contributions include:

- We present a novel technique for OOD detection, which computes the angles between the feature representation and its projection to the decision boundaries, relative to the mean of ID-features

- Our score is model agnostic, hyperparameter-free and efficient, scaling linearly with the number of ID-classes. Therefore, it can flexibly be combined with various architectures without the need of additional tuning.

- We demonstrate the state-of-the-art performance of LAFO on widely used CIFAR-10 and ImageNet OOD benchmarks. Specifically, LAFO achieves a $7.74\%$ reduction in FPR95 score compared to the best previous distance-based method on a large-scale ImageNet OOD benchmark.

- The scale-invariant property of LAFO allows for straightforward aggregation of confidence scores from multiple pre-trained models, improving ensemble performance. Our experiments show that the ensemble with LAFO reduces the FPR95 by $2.51\%$ on the ImageNet OOD benchmark compared to the best single model performance.

## 2 RELATED WORK

Previous work in OOD detection falls into two categories: (i) methods that regularize models during training to produce different outcomes for ID and OOD data, and (ii) post-hoc methods that develop scoring mechanisms using pre-trained models on ID data.

**Model Regularization**   Early methods addressing the OOD detection problem **?**Hendrycks et al.; Mohseni et al. (2020); Yang et al. (2021) utilize additional datasets to represent out-of-distribution data, training models with both positive and negative samples. This approach assumes a specific nature of OOD data, potentially limiting its effectiveness when encountering OOD samples that deviate from this assumption during inference. Malinin & Gales (2018) designed a network architecture to measure distributional uncertainty using Dirichlet Prior Networks. In Geifman & El-Yaniv (2019)'s work, they provide another architecture with an additional reject option to abstain from answering. Their selection model incorporates a hyperparameter, the coverage rate for ID, to control the percentage of ID samples classified. Lee et al. (2018a); Ming et al. (2022); Du et al., focused on synthesizing outliers rather than relying on auxiliary datasets to improve the generalizability of the detection method. On the other hand, Meinke & Hein; Van Amersfoort et al. (2020); Wei et al. (2022) argued that overconfident predictions of the networks on OOD data are the problem to be mitigated. For example, Van Amersfoort et al. (2020) puts an additional gradient penalty to limit the confidence of the network. Whereas, Wei et al. (2022) tackled the same problem by enforcing a constant logit vector norm during training. Although it is natural to impose structures during training for better separability of ID and OOD, these methods face the trade-off between OOD separability and model performance. Moreover, such approaches lack the flexibility of post-hoc score functions, as they necessitate model retraining—a process that can be both time-consuming and computationally expensive.

**Score Functions**   Recently, developing score functions for pretrained models on ID data has gained attention due to its ease of implementation and flexibility. These methods typically either couple feature representations with distance metrics, or measure a model's confidence using its logits. Beyond canonical works such as Maximum Softmax Probability (Hendrycks & Gimpel, 2022), ODIN score

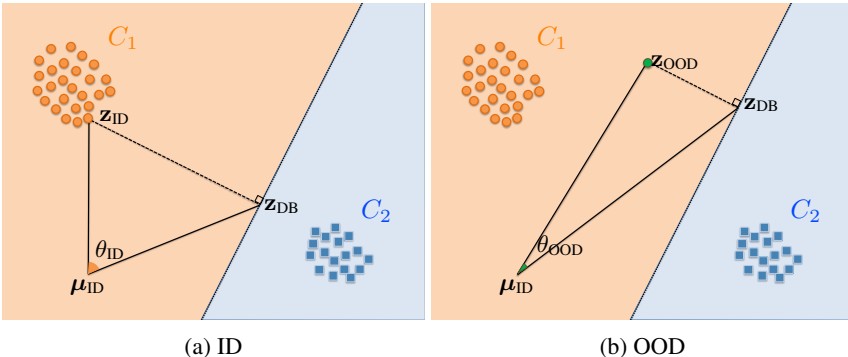

Figure 1: Geometric visualization of LAFO for in-distribution (*left*) and out-of-distribution (*right*) cases. LAFO focuses on the angular distance between the feature representation and the decision boundary, from the perspective of the in-distribution mean. The angle $\theta$ serves as the distinguishing factor between ID and OOD samples, with $\theta_{\text{ID}} > \theta_{\text{OOD}}$.

(Liang et al., 2018), Energy score (Liu et al., 2020), Mahalanobis score (Lee et al., 2018b), Virtual Logit Matching Wang et al. (2022), we observed many advancements in post-hoc score design. For example, the activation shaping algorithms such as ASH (Djurisic et al., 2023), Scale (Xu et al.), and ReAct (Sun et al., 2021), apply activation truncations to feature representations, reducing model's confidence for OOD data. These methods can be used in conjunction with LAFO improving the performance. Following a similar intuition, DICE Sun & Li (2022) applies weight sparsification to limit the model confidence by using the more salient weights for output. GradOrth Behpour et al. (2023) offers a gradient-based perspective, projecting representations to a lower dimensional subspace based on gradient norms. Recent distance-based methods KNN Sun et al. (2022) and FDBD Liu & Qin (2024) successfully utilized the feature representations from networks trained with supervised contrastive loss. KNN assigns a score to a sample based on the kth nearest neighbor in ID training data. On the other hand, FDBD assigns a score to a sample based on its estimate of the distance between the feature representation and the decision boundaries.

Our work falls into the score function category, serving as a plug-in for any pre-trained model on ID data. LAFO combines feature space and logit space methods by utilizing the relative angle between the feature representation and its projections to the decision boundaries. Among the existing works, the closest approach to our method is Liu & Qin (2024), which uses a lower bound estimate to the decision boundaries. However, the regularization term they introduced to equalize deviations from the in-distribution mean inadvertently includes a term in their equation that is uncorrelated with being out- or in-distribution and can change spuriously, impeding performance. In contrast, we provide a simple, hyperparameter-free score function that effectively incorporates in-distribution context and maintains scale invariance, all without extra regularization terms.

## 3 METHOD

### 3.1 PROBLEM SETTING

We consider a supervised classification setting with input space $\mathcal{X}$ and label space $\mathcal{Y}$, following the literature Yang et al. (2024). Given a model $f : \mathcal{X} \to \mathbb{R}^{|\mathcal{Y}|}$ pretrained on an in-distribution dataset $D_{\text{ID}} = \{(\mathbf{x}_i, y_i)\}_{i=1}^N$, where elements of $D_{\text{ID}}$ are drawn from a joint distribution $P_{\mathcal{X}\mathcal{Y}}$, with support $\mathcal{X} \times \mathcal{Y}$. We denote with $P_{\text{ID}}$ its marginalization on $\mathcal{X}$. The *OOD detection problem* aims to determine whether an input sample originates from the in-distribution $P_{in}$ or not. Denoting with $\mathcal{Y}_{\text{OOD}}$ a set of labels such that $\mathcal{Y} \cap \mathcal{Y}_{\text{OOD}} = \emptyset$, OOD samples are drawn from a distribution $P_{\text{OOD}}$ which correspond to the marginalization on $\mathcal{X}$ of the joint distribution on $\mathcal{X} \times \mathcal{Y}_{\text{OOD}}$. i.e., they share the same input space $\mathcal{X}$ as in-distribution samples, but have labels outside $\mathcal{Y}$. Shifting from $P_{\text{ID}}$ to $P_{\text{OOD}}$ corresponds to a semantic change in label space.

---

**Algorithm 1** LAFO (Look Around and Find Out)

---

**Require:** Sample $\mathbf{x}$, Pretrained model $f$, Mean of the in-distribution features $\boldsymbol{\mu}_{\text{ID}}$
**Ensure:** OOD score $s$
 1: **function** LAFO($\mathbf{x}, f, \boldsymbol{\mu}_{\text{ID}}, \alpha$)
 2:  $\hat{y} \leftarrow \arg\max_{y \in \mathcal{Y}} f(\mathbf{x})$
 3:  $\mathbf{z} = f_1 \circ \ldots \circ f_{L-1}(\mathbf{x})$       ▷ Compute the penultimate last layer features $\mathbf{z}$
 4:  $score \leftarrow -\infty$
 5:  **for** $y' \in \mathcal{Y}$ **and** $y' \neq \hat{y}$ **do**               ▷ For each other class
 6:    compute $\mathbf{z}_{db}$ as in Eq.1
 7:    compute $\theta_{\hat{y},y'}(\mathbf{z})$ using Eq. 2
 8:    compute $\tilde{s}(\mathbf{z})$ using Eq. 3
 9:    **if** $\tilde{s}(\mathbf{z}) \geq score$ **then**
10:      $score = \tilde{s}(\mathbf{z})$
11:    **end if**
12:  **end for**
13:  **return** $score$       ▷ Returns the maximum score across all other classes
14: **end function**

---

The OOD decision can be made via the function $d : \mathcal{X} \rightarrow \{\text{ID}, \text{OOD}\}$ given a *score function* $s : \mathcal{X} \rightarrow \mathbb{R}$ such that:

$$d(\mathbf{x}; s, f) = \begin{cases} \text{ID} & \text{if } s(\mathbf{x}; f) \geq \lambda \\ \text{OOD} & \text{if } s(\mathbf{x}; f) < \lambda \end{cases}$$

where samples with high scores are classified as in-distribution, according to the threshold $\lambda$. For example, to compute the standard FPR95 metric (Yang et al., 2024), the threshold $\lambda$ is chosen such that it correctly classifies 95% of ID held-out data. An ideal OOD score function should capture differences in model outputs between samples drawn from $P_{\text{ID}}$ and $P_{\text{OOD}}$, effectively, determining when the model encounters inputs from classes it was not trained on.

### 3.2 OOD DETECTION WITH RELATIVE ANGLES

This section presents our OOD score function, which relates the feature representations with the decision boundaries using relative angles in the feature space to discriminate between ID and OOD samples. Figure 1 provides a geometric visualization of our method. Our approach leverages the geometric relationships between three key points in the feature space: (i) the initial representation of a sample, (ii) its projection onto the decision boundary, and (iii) the mean of in-distribution features.

We propose using the relation between feature representations and decision boundaries by deriving closed-form plane equations for the decision boundaries between any two classes. Specifically, we examine the angle formed between the feature representation vector and its projection onto the decision boundary. However, this angle is sensitive to the choice of origin, creating an ambiguity as the geometric relationship between the feature representation and the decision boundary should be translation-invariant. To address this, we propose to represent features in a reference frame relative to the mean of the in-distribution samples. Therefore, we incorporate in-distribution characteristics by centering around its mean, while ensuring scale and translation invariance.

We observe that the angle between the centered representation and its projection onto the decision boundary is larger for ID data, indicating them requiring higher cost to change their label which captures the model's confidence. In contrast, for OOD data, angle is smaller since they are expected to be more unstable, as they do not contain strong clues about their predicted classes (see Figure 1 for a conceptual explanation and the ID/OOD histograms in Appendix A.1 for empirical evidence).

Our framework provides a concise scoring with useful properties such as translation and scale invariance. These properties enable LAFO to be used in conjunction with existing activation shaping algorithms and allow for confidence aggregation across different models through score summation.

### 3.3 Features on the Decision Boundary

In this section, we derive the mathematical equations and demonstrate the properties of our score. The model $f$ can be rewritten as a composed function $f_1 \circ ... f_{L-1} \circ g$, where $L$ is the number of layers and $g : \mathbb{R}^D \to \mathbb{R}^{|\mathcal{Y}|}$ corresponds to the last layer classification head. The function $g(\mathbf{z}) = \mathbf{W}\mathbf{z} + \mathbf{b}$ maps penultimate layer features $\mathbf{z} \in \mathbb{R}^D$ to the logits space via $\mathbf{W} \in \mathbb{R}^{|\mathcal{Y}| \times D}$ and $\mathbf{b} \in \mathbb{R}^{|\mathcal{Y}|}$. The decision boundary between any two classes $y_1$ and $y_2$ with $y_1 \neq y_2$ can be represented as:

$$DB_{y_1,y_2} = (\mathbf{w}_{y_1} - \mathbf{w}_{y_2})^T \mathbf{z} + b_{y_1} - b_{y_2} = 0$$

where $\mathbf{w}_{y_1}$ (or $\mathbf{w}_{y_2}$) denotes the the row vectors of $\mathbf{W}$ corresponding to class $y_1$ (respectively $y_2$) and similarly, $b_{y_1}, b_{y_2}$ are the bias values corresponding to classes $y_1$ and $y_2$. Intuitively, given a fixed classifier, this equation is satisfied for all $\mathbf{z}$'s such that their corresponding logits for class $y_1$ and $y_2$ are equal. Then, feature representations can be projected orthogonally onto the hyperplane that defines the decision boundary:

$$\mathbf{z}_{db} = \mathbf{z} - \frac{(\mathbf{w}_{y_1} - \mathbf{w}_{y_2})^T \mathbf{z} + (b_{y_1} - b_{y_2})}{||\mathbf{w}_{y_1} - \mathbf{w}_{y_2}||^2}(\mathbf{w}_{y_1} - \mathbf{w}_{y_2}) \tag{1}$$

Let $\boldsymbol{\mu}_{\text{ID}} \in \mathbb{R}^D$ be the mean of the in-distribution feature representations. Centering w.r.t. $\boldsymbol{\mu}_{\text{ID}}$ corresponds to shifting the origin to $\mu_{\text{ID}}$. In this new reference frame, three key points form a triangle in $D$-dimensional space: the centered feature vector $(\mathbf{z} - \boldsymbol{\mu}_{\text{ID}})$, its projection onto the decision boundary $(\mathbf{z}_{\mathbf{db}} - \boldsymbol{\mu}_{\text{ID}})$ and the new origin (see Figure 1). Then, rather than the absolute distance between $\mathbf{z}$ and $\mathbf{z}_{db}$, we consider the relative angle $\theta_{y_1,y_2}(\mathbf{z})$ from the in-distribution feature representation's reference frame: this captures *the position of features and the decision boundaries with respect to the in-distribution data*, while also being scale invariant:

$$\theta_{y_1,y_2}(\mathbf{z}) = \arccos\left(\frac{< \mathbf{z} - \boldsymbol{\mu}_{\text{ID}}, \mathbf{z}_{db} - \boldsymbol{\mu}_{\text{ID}} >}{||\mathbf{z} - \boldsymbol{\mu}_{\text{ID}}|| \cdot ||\mathbf{z}_{db} - \boldsymbol{\mu}_{\text{ID}}||}\right) \tag{2}$$

Our score function extracts the maximum discrepancy of the relative angles between the centered feature representation and its projections on $DB_{\hat{y},y'}$, where $\hat{y} = \arg\max_{y \in \mathcal{Y}} g(\mathbf{z})$ and $y' \in \mathcal{Y}$, $y' \neq y$. Therefore for a sample $\mathbf{x} \in \mathcal{X}$, given $\mathbf{z} = f_1 \circ ... \circ f_{L_1}(\mathbf{x})$ we can write the score $s(\mathbf{x}, f)$ as a function of $\mathbf{z}$:

$$\tilde{s}(\mathbf{z}) = \max\left(\{\theta_{y,y'}(\mathbf{z})\}_{y' \in \mathcal{Y}, y' \neq y}\right) \tag{3}$$

**Properties.** Intuitively, our score function captures several key aspects:

- **Confidence Measure.** The angle between the feature representation and its projection onto a decision boundary is proportional to the distance between them, serving as a proxy for the model's confidence.
- **In-Distribution Context.** By centering the space using the mean of in-distribution features, we incorporate ID statistics, improving angle separability across points.
- **Maximum Discrepancy.** By considering the maximum angle across all non-predicted classes, we identify the 'furthest' relative decision boundary, extracting model's confidence on least likely class that an instance might belong.
- **Scale Invariance.** Unlike absolute distances, angles remain consistent even if the feature space is scaled, allowing for fair comparisons between different models.

**Relation with the state-of-the-art fDBD (Liu & Qin, 2024)).** We now provide a geometric interpretation for the score function fDBD. Using our analysis, we identified that their score can directly be mapped into the triangle we formed in Figure 1. For a sample $\mathbf{x} \in \mathcal{X}$:

$$\text{fDBD}(\mathbf{z}) = \frac{d(\mathbf{z}, \mathbf{z}_{db})}{||\mathbf{z} - \boldsymbol{\mu}_{\text{ID}}||_2}$$

where $\mathbf{z} \in \mathbb{R}^D$ is the feature representations of the input $x \in \mathcal{X}$, $\mathbf{z}_{db} \in \mathbb{R}^D$ is its projection onto the decision boundary, and $d(\cdot, \cdot)$ is the euclidean distance. Although seemingly unrelated, we can connect this score to our relative angle and demonstrate that the regularization term on the denominator brings a term that does not effectively discriminate between OOD and ID. As a result, impeding fDBD's performance.

Using translation invariance of the euclidean distance, the same score can be written as:

$$\text{fDBD}(\mathbf{z}) = \frac{d(\mathbf{z} - \boldsymbol{\mu}_{\text{ID}}, \mathbf{z}_{db} - \boldsymbol{\mu}_{\text{ID}})}{d(\mathbf{z} - \boldsymbol{\mu}_{\text{ID}}, 0)}$$

One can observe that, this is the ratio of two sides of the triangle formed between the points $\mathbf{z} - \boldsymbol{\mu}_{\text{ID}}$, $\mathbf{z_{db}} - \boldsymbol{\mu}_{\text{ID}}$ and the origin. Using the law of sines:

$$\frac{d(\mathbf{z} - \boldsymbol{\mu}_{\text{ID}}, \mathbf{z}_{db} - \boldsymbol{\mu}_{\text{ID}})}{\sin(\theta)} = \frac{d(\mathbf{z} - \boldsymbol{\mu}_{\text{ID}}, 0)}{\sin(\alpha)} \tag{4}$$

$$\frac{\sin(\theta)}{\sin(\alpha)} = \frac{d(\mathbf{z} - \boldsymbol{\mu}_{\text{ID}}, \mathbf{z}_{db} - \boldsymbol{\mu}_{\text{ID}})}{d(\mathbf{z} - \boldsymbol{\mu}_{\text{ID}}, 0)} = \text{fDBD}(\mathbf{z}) \tag{5}$$

where $\theta$ and $\alpha$ are the angles opposite to the sides $\mathbf{z} - \boldsymbol{\mu}_{\text{ID}} - (\mathbf{z_{db}} - \boldsymbol{\mu}_{\text{ID}})$ and $\mathbf{z} - \boldsymbol{\mu}_{\text{ID}}$ respectively. Although the observation they made on comparing the distances to the decision boundaries at equal deviation levels from the mean of in-distribution is inspiring, we claim that the angle $\alpha$ is not very informative for ID and OOD separation. This is because $\alpha$ is connected to the magnitude of the feature vector relative to $\boldsymbol{\mu}_{\text{ID}}$, which may not directly correlate with OOD characteristics. On Figure 2 we show the $\sin(\alpha)$ values between CIFAR-10 (Krizhevsky et al., 2009) and Texture (Cimpoi et al., 2014) datasets, empirically justifying that including this term impedes fDBD's performance. Omitting the denominator from Equation 5 allows to effectively capture the relation between the feature representation and the decision boundary from the mean of in-distribution's view.

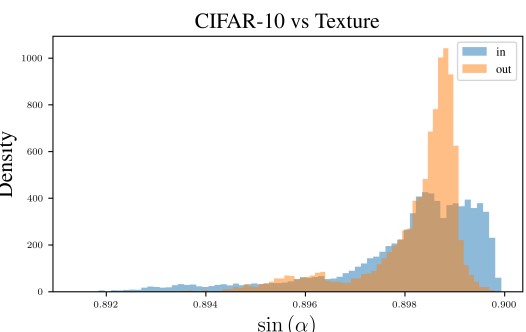

Figure 2: Histogram of ID (CIFAR-10) and OOD (Texture) with respect to the sine of the angle of the triangle that looks at the edge $\mathbf{z} - \boldsymbol{\mu}_{\text{ID}}$. This empirically shows that $\sin(\alpha)$ is not highly informative to distinguish between ID or OOD.

## 4 EXPERIMENTS

In this section, we demonstrate the performance of LAFO on various settings and benchmarks. We will first provide our results on the most common benchmarks CIFAR-10 (Krizhevsky et al., 2009) and ImageNet (Deng et al., 2009) to show small and large scale performance of our scoring. Then, we show the flexibility of LAFO by (i) combining it with different activation shaping algorithms, (ii) using it to aggregate different architectures' confidences via simply summing their scores, exploiting the scale-invariance property of LAFO. Moreover, an ablation study is presented to show the efficacy of the design choices.

**Benchmarks:** We mainly consider two widely used benchmarks: CIFAR-10 (Krizhevsky et al., 2009) and ImageNet (Deng et al., 2009)). We included the evaluation on CIFAR-10 OOD

Table 1: LAFO achieves state-of-the-art performance on CIFAR-10 OOD benchmark. Evaluated on ResNet-18 with FPR95 and AUROC. ↑ indicates that larger values are better and vice versa. Best performance highlighted in **bold**. Methods with * are hyperparameter-free.

| Method | SVHN | | iSUN | | Place365 | | Texture | | Avg | |
|---|---|---|---|---|---|---|---|---|---|---|
| | FPR95↓ | AUROC↑ | FPR95↓ | AUROC↑ | FPR95↓ | AUROC↑ | FPR95↓ | AUROC↑ | FPR95↓ | AUROC↑ |
| MSP * | 59.51 | 91.29 | 54.57 | 92.12 | 62.55 | 88.63 | 66.49 | 88.50 | 60.78 | 90.14 |
| ODIN | 61.71 | 89.12 | 15.09 | 97.37 | 41.45 | 91.85 | 52.62 | 89.41 | 42.72 | 91.94 |
| Energy * | 53.96 | 91.32 | 27.52 | 95.59 | 42.80 | 91.03 | 55.23 | 89.37 | 44.88 | 91.83 |
| ViM | 25.38 | 95.40 | 30.52 | 95.10 | 47.36 | 90.68 | 25.69 | 95.01 | 32.24 | 94.05 |
| MDS | 16.77 | 95.67 | 7.56 | 97.93 | 85.87 | 68.44 | 35.21 | 85.90 | 36.35 | 86.99 |
| CSI | 37.38 | 94.69 | 10.36 | 98.01 | 38.31 | 93.04 | 28.85 | 94.87 | 28.73 | 95.15 |
| SSD+ | 1.35 | 99.72 | 33.60 | 95.16 | 26.09 | 95.48 | 12.98 | 97.70 | 18.51 | 97.02 |
| ReAct | 6.15 | 98.75 | 10.31 | 98.09 | 21.68 | 95.47 | 10.18 | 98.12 | 12.08 | 97.61 |
| KNN+ | 2.20 | 99.57 | 20.06 | 96.74 | 23.06 | 95.36 | 8.09 | 98.56 | 13.35 | 97.56 |
| fDBD * | 4.59 | 99.00 | 10.04 | 98.07 | 23.16 | 95.09 | 9.61 | 98.22 | 11.85 | 97.60 |
| LAFO * | 3.53 | 99.16 | 8.36 | 98.28 | 23.40 | 94.88 | 8.58 | 98.34 | **10.97** | **97.67** |

Benchmark to show the performance on smaller scale datasets. In CIFAR-10 experiments, we use a pretrained ResNet-18 architecture He et al. (2016) trained with supervised contrastive loss (Khosla et al., 2020), following previous literature Liu & Qin (2024); Sun et al. (2022); Sehwag et al. (2021). During inference 10.000 test samples are used to set the in-distribution scores and choose the threshold value $\lambda$; while the datasets SVHN (Netzer et al., 2011), iSUN (Xu et al., 2015), Places365 (Zhou et al., 2017) and Texture (Cimpoi et al., 2014) are used to obtain out-of-distribution scores and metric evaluation. Similarly, for large scale ImageNet OOD Benchmark, we use a pretrained ResNet-50 architecture trained with supervised contrastive loss. In-distribution validation set of size 50.000 is used to set the ID scores and the threshold, and the datasets iNaturalist (Van Horn et al., 2018), SUN (Xiao et al., 2010), Places365 (Zhou et al., 2017) and Texture (Cimpoi et al., 2014) are used to obtain out-of-distribution scores and metric evaluation. We analyzed the case of architectures trained with Cross Entropy loss, in the ensemble experiment of Table 3. Note that, in this typical OOD Detection Benchmarks the samples that have same classes as ID are removed from their OOD counterparts, following the work Huang & Li (2021) and fitting into our problem setting.

**Metrics:** Throughout our experiments we report two metrics: False-positive rate at $\%95$ true positive rate (FPR95), and Area Under the Receiver Operating Characteristic Curve (AUROC). FPR95, simply measures what percentage of OOD data we falsely classify as ID where our threshold includes $95\%$ of ID data. Therefore, smaller FPR95 indicates a better performance by sharply controlling the false positive rate. On the other hand, AUROC measures the model's ability to distinguish between ID and OOD by calculating the area under the curve that plots the true positive rate against the false positive rate across various thresholds. AUROC shows how rapidly we include ID data while paying the cost of including false positives. Thus, higher AUROC shows a better result.

## 4.1 OOD DETECTION ON CIFAR-10 AND IMAGENET BENCHMARKS

Table 1 and Table 2 shows the performance of LAFO along with the 9 baselines on CIFAR-10 and ImageNet OOD Benchmarks, respectively. All the baselines on CIFAR-10 use ResNet-18 architecture and on ImageNet use ResNet-50. We compare with methods in three categories : (i) logit-based score functions, (ii) methods that utilize contrastively learned representations (iii) hyperparameter free methods. Our proposed method reaches *state-of-the-art* performance on both benchmarks, reducing the FPR95 on average by $7.74\%$ on Imagenet and $0.88\%$ on CIFAR10. In the following, we provide a detailed analysis of these results.

**LAFO continues to show the success of distance-based methods over logit-based methods.** Logit-based scoring methods MSP Hendrycks & Gimpel (2022), Energy Liu et al. (2020) are one of the earliest baselines proving their success on measuring model's confidences. MSP measures the maximum softmax probability as its score while Energy does a logsumexp operation on the logits. Recent distance-based methods like KNN+ Sun et al. (2022) and fDBD Liu & Qin (2024) outperforms the early logit-based ones. Similarly, LAFO achieves significantly better performance on both benchmarks, reducing the FPR95 up to $49.81\%$ and $41.91\%$ while improving the AUROC up to $7.53\%$ and $12.27\%$ on CIFAR-10 and ImageNet OOD benchmarks.

Table 2: LAFO achieves state-of-the-art performance on ImageNet OOD benchmark. Evaluated on ResNet-50 with FPR95 and AUROC. ↑ indicates that larger values are better and vice versa. Best performance highlighted in **bold**. Methods with * are hyperparameter-free.

| Method | iNaturalist | | SUN | | Places | | Texture | | Avg | |
|---|---|---|---|---|---|---|---|---|---|---|
| | FPR95↓ | AUROC↑ | FPR95↓ | AUROC↑ | FPR95↓ | AUROC↑ | FPR95↓ | AUROC↑ | FPR95↓ | AUROC↑ |
| MSP * | 54.99 | 87.74 | 70.83 | 80.63 | 73.99 | 79.76 | 68.00 | 79.61 | 66.95 | 81.99 |
| ODIN | 47.66 | 89.66 | 60.15 | 84.59 | 67.90 | 81.78 | 50.23 | 85.62 | 56.48 | 85.41 |
| Energy * | 55.72 | 89.95 | 59.26 | 85.89 | 64.92 | 82.86 | 53.72 | 85.99 | 58.41 | 86.17 |
| ViM | 71.85 | 87.42 | 81.79 | 81.07 | 83.12 | 78.40 | 14.88 | 96.83 | 62.91 | 85.93 |
| MDS | 97.00 | 52.65 | 98.50 | 42.41 | 98.40 | 41.79 | 55.80 | 85.01 | 87.43 | 55.17 |
| SSD+ | 57.16 | 87.77 | 78.23 | 73.10 | 81.19 | 70.97 | 36.37 | 88.53 | 63.24 | 80.09 |
| ReAct | 20.38 | 96.22 | 24.20 | 94.20 | 33.85 | 91.58 | 47.30 | 89.80 | 31.43 | 92.95 |
| KNN+ | 30.18 | 94.89 | 48.99 | 88.63 | 59.15 | 84.71 | 15.55 | 95.40 | 38.47 | 90.91 |
| fDBD * | 17.27 | 96.68 | 42.30 | 90.90 | 49.77 | 88.36 | 21.83 | 95.43 | 32.78 | 92.86 |
| LAFO * | 12.27 | 97.42 | 31.80 | 92.85 | 40.71 | 90.10 | 15.39 | 96.68 | **25.04** | **94.26** |

**LAFO improves on the recent success of methods using contrastively learned features.** Table 1 and 2 show the success of recent methods CSI Tack et al. (2020), SSD+ Sehwag et al. (2021), KNN+ Sun et al. (2022) and fDBD Liu & Qin (2024) that utilizes contrastively learned representations over the ones those do not use. We observe that the additional structure the supervised contrastive loss puts on the feature representations are particularly beneficial to the distance-based methods. LAFO also benefits from more structured representations on the feature space, as it explores the relationship between the representation and the decision boundaries. Notably, LAFO improves both of the metrics on both CIFAR-10 and ImageNet benchmarks, achieving the state-of-the-art performance.

**LAFO enjoys being hyperparameter-free while offering state-of-the-art performance.** Methods that use hyperparameters require a holdout set the tune them. Moreover, having different optimal hyperparameters for different benchmarks makes it harder to use them in real world applications. MSP Hendrycks & Gimpel (2022), Energy Liu et al. (2020) and fDBD Liu & Qin (2024) are the baselines which do not require any hyperparameters. LAFO outperforms the most competitive hyperparameter-free baseline fDBD by having 10.97% FPR95 on CIFAR-10 as opposed to fDBD's 11.85%, and 25.04% FPR95 on ImageNet compared to fDBD's 32.78%.

### 4.2 MODEL ENSEMBLING WITH LAFO

Recent works Xue et al. (2024) and Xu et al. (2024) show that creating an ensemble of models can enhance the OOD performance. Inspired from these works, and from the observation that scale invariant representations are compatible between distinct models (Moschella et al.), we demonstrate that *scale-invariant score functions can aggregate the confidences from different models*, by simply summing their scores. On Table 3 we show the individual performances of models ResNet-50, ResNet-50 with supervised contrastive loss and ViT-B/16 as well as their combined performances using the scale-invariant LAFO.

Note that, we show the scale-invariance property of fDBD on section 3.3 and added it to demonstrate the same idea and also to compare with LAFO. It can be seen that for both of the score functions, the performance of ensemble is better than their individual counterparts showing that score aggregation improves their OOD performance. Moreover, the ensemble with LAFO achieves a performance with 22.53% FPR95 and 96.41% AUROC, improving the metrics compared to the best individual performer in the ensemble by 2.51% and 2.15% respectively. We show qualitatve evidence of the improved performance by plotting the ID and OOD histograms on ImageNet

Figure 3: LAFO can be used for ensemble OOD detection due to its scale-invariance property. Evaluated on ImageNet OOD benchmark. Best performance highlighted in **bold**.

| Method | Avg | |
|---|---|---|
| | FPR95↓ | AUROC↑ |
| fDBD w/ResNet50 | 51.35 | 89.20 |
| fDBD w/ResNet50-supcon | 32.78 | 92.86 |
| fDBD w/ViT-B/16 | 41.55 | 91.05 |
| LAFO w/ResNet50 | 44.58 | 90.68 |
| LAFO w/ResNet50-supcon | 25.04 | 94.26 |
| LAFO w/ViT-B/16 | 39.92 | 91.38 |
| Ensemble fDBD | 31.05 | 95.29 |
| Ensemble LAFO | **22.53** | **96.41** |

(Deng et al., 2009)(ID) and iNaturalist (Van Horn et al., 2018) (OOD) datasets, respectively, in Figure 6 in the Appendix, demonstrating a better separation in the ensembled model. In summary,

we demonstrate that scale-invariance of LAFO allows aggregating different models' confidences to solve OOD Detection Problem.

Table 3: LAFO can be used as a plug-in on top of activation shaping algorithms. Evaluated under ImageNet OOD benchmark. ↑ indicates that larger values are better and vice versa. Best performance highlighted in **bold**.

| Method | iNaturalist | | SUN | | Places | | Texture | | Avg | |
|---|---|---|---|---|---|---|---|---|---|---|
| | FPR95↓ | AUROC↑ | FPR95↓ | AUROC↑ | FPR95↓ | AUROC↑ | FPR95↓ | AUROC↑ | FPR95↓ | AUROC↑ |
| LAFO w/ReLU | 12.27 | 97.42 | 31.80 | 92.85 | 40.71 | 90.10 | 15.39 | 96.68 | 25.04 | 94.26 |
| LAFO w/ASH | 11.08 | 97.68 | 27.81 | 93.59 | 36.53 | 91.36 | 18.48 | 96.70 | 23.47 | 94.58 |
| LAFO w/Scale | 14.65 | 97.05 | 25.43 | 94.02 | 36.21 | 90.78 | 17.07 | 95.65 | 23.34 | 94.37 |
| LAFO w/ReAct | 11.13 | 97.79 | 22.34 | 94.95 | 33.33 | 91.81 | 14.65 | 96.60 | **20.36** | **96.29** |

## 4.3 LAFO WITH ACTIVATION SHAPING ALGORITHMS

Recent methods ReAct Sun et al. (2021), ASH Djurisic et al. (2023) and Scale Xu et al. show their success to modify the feature representations to reduce model's overconfident predictions. All three methods adopt a hyperparameter percentile to choose how to truncate and scale the feature representations using ID data statistics. When combined with Energy Liu et al. (2020) score, these methods improve the OOD Detection performance. On Table 15 we show that applying LAFO scoring after activation shaping algorithms improves the performance. Specifically combining LAFO with ReAct reduces FPR95 from $25.04\%$ to $20.36\%$ highlighting both the flexibility and efficacy of our method. This demonstrates that LAFO can flexibly be combined with activation shaping algorithms.

## 4.4 ABLATION STUDIES

In this section, we will demonstrate the effectiveness of design choices on our score function LAFO. We first justify our choice of centering in $\boldsymbol{\mu}_{\text{ID}}$ empirically, among the candidates: $\boldsymbol{\mu}_{\text{ID}}$, $\boldsymbol{\mu}_{y_{\text{pred}}}$, $\boldsymbol{\mu}_{y_{\text{target}}}$ and $\max(\mathbf{z}_{\text{ID}})$. Then, we compare different angle aggregation techniques across classes by replacing our $\max(\{\theta_y, y'\}_{y' \in \mathcal{Y}, y' \neq y})$ with mean and min across classes.

Table 4: Ablation on the different origin perspectives for centering. Evaluated under both ImageNet and CIFAR-10 OOD benchmarks.

| Method | CIFAR-10 | | ImageNet | |
|---|---|---|---|---|
| | FPR95↓ | AUROC↑ | FPR95↓ | AUROC↑ |
| LAFO w/ $\mu_{y_{\text{pred}}}$ | 12.42 | 97.59 | 43.02 | 89.86 |
| LAFO w/ $\mu_{y_{\text{target}}}$ | 13.26 | 97.48 | 28.29 | 93.46 |
| LAFO w/ $\max(\mathbf{z}_{\text{ID}})$ | 13.39 | 97.42 | 32.44 | 92.01 |
| LAFO w/ $\mu_{\text{ID}}$ | **10.97** | **97.67** | **25.04** | **94.26** |

Table 5: Ablation on the different score aggregations across classes. Evaluated under both ImageNet and CIFAR-10 OOD benchmarks.

| Method | CIFAR-10 | | ImageNet | |
|---|---|---|---|---|
| | FPR95↓ | AUROC↑ | FPR95↓ | AUROC↑ |
| LAFO w/min | 32.02 | 95.23 | 79.15 | 81.38 |
| LAFO w/mean | 11.84 | 97.59 | 32.76 | 92.87 |
| LAFO w/max | **10.97** | **97.67** | **25.04** | **94.26** |

**Centering with $\mu_{\text{ID}}$ incorporates ID-statistics without biasing towards one particular class.** Table 4 shows the performance comparison between centerings with respect to different points. Using the relative angle with respect to the predicted ($\boldsymbol{\mu}_{y_{\text{pred}}}$) or target ($\boldsymbol{\mu}_{y_{\text{target}}}$) class centroid induce a bias towards the corresponding class, which in the end hinders the compatibility between angles coming across classes. On the other hand, using $\max(\mathbf{z}_{\text{ID}})$ shifts every feature representation to the same orthant, reducing to simply computing the absolute distance between feature representations and the decision boundaries, which is agnostic from the in-distribution feature statistics. We observe a significant improvement in performance when computing relative angles using $\mu_{\text{ID}}$, demonstrating the importance of incorporating in-distribution (ID) statistics when measuring the relationship between feature representations and decision boundaries. LAFO with $\mu_{\text{ID}}$ centering improves the FPR95 by up to $1.45\%$ and $7.4\%$ on CIFAR-10 and ImageNet respectively while also improving the AUROC for both benchmarks.

**Looking at the furthest class is better for ID/OOD separation.** On Table 5 we explored different ways to aggregate class specific angles. Originally, we devise our score function to return the maximum relative angle discrepancy between the feature representation across decision boundaries. Intuitively, this suggests that considering the furthest possible class that a feature belongs from the mean of in-distribution's perspective is effective to distinguish OOD from ID. On the other hand,

comparing the minimum focuses on the smallest relative angle, reducing the separability significantly. Table 5 demonstrates taking the maximum across classes clearly outperforms mean and min aggregations, improving FPR95 and AUROC metrics on both benchmarks. Specifically the difference is higher on our large-scale experiments reducing the FPR95 by $7.72\%$ and increasing the AUROC by $1.39\%$ compared to the second best aggregation.

## 5 CONCLUSION

In this paper, we introduce a novel angle-based OOD detection score function. As a post-hoc measure of model confidence, LAFO offers several key advantages: it is (i) hyperparameter-free, (ii) model-agnostic and (iii) scale-invariant. These features allow LAFO to be applied to arbitrary pretrained models and used in conjunction with existing activation shaping algorithms, enhancing the performance. Notably, its scale-invariant nature enables simple aggregation of multiple models' confidences through score summation, allowing a creation of an effective model ensemble for OOD detection. Our extensive experiments demonstrate that LAFO achieves state-of-the-art performance, using the relationship between the feature representations and decision boundaries relative to the ID statistics effectively. Despite the state-of-the-art performance achieved by LAFO, one possible limitation might be the use of the mean alone to capture the ID statistics in our score. As a future direction, we plan to mitigate this possible limitation by incorporating multiple relative angles to better capture the ID-statistics beyond the mean, aiming to further improve OOD detection performance. We hope that our approach inspires further research into geometric interpretations of model confidence for OOD detection.

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

# A APPENDIX

## A.1 COMPARISON OF SCORE DISTRIBUTIONS: CIFAR-10 VS. OOD DATASETS

In this section we report the score histogram results on the Table 1 for CIFAR-10 OOD Benchmark.

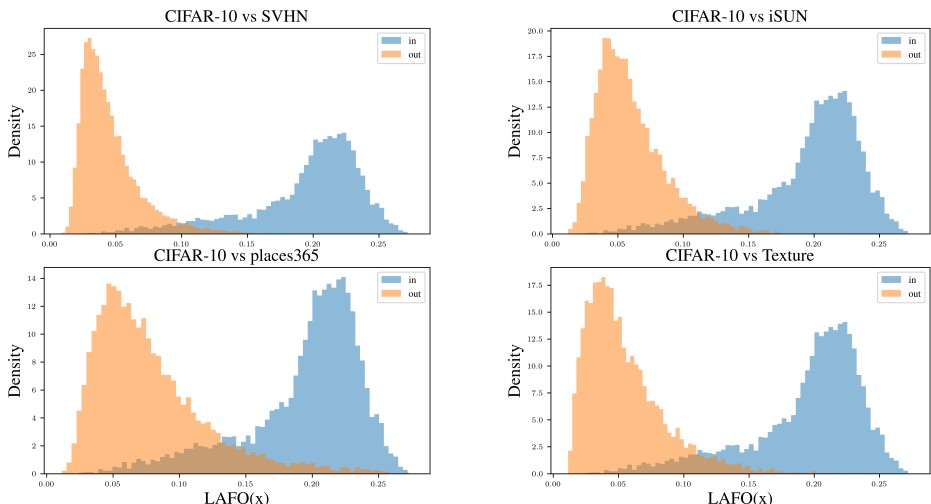

Figure 4: Score distributions of ID and OOD datasets in CIFAR-10 OOD Benchmark.

## A.2 COMPARISON OF SCORE DISTRIBUTIONS: IMAGENET VS. OOD DATASETS

We report the score histogram results on the Table 2 for ImageNet OOD Benchmark.

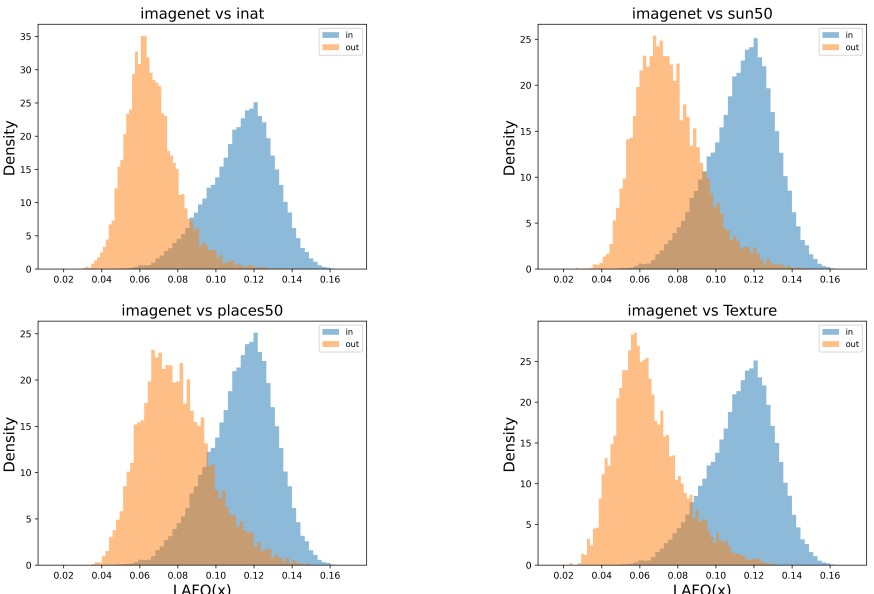

Figure 5: Score distributions of ID and OOD datasets in ImageNet OOD Benchmark.

Table 6: LAFO can be used as a score function to accumulate different architectures' confidences due to its scale-invariance property. Evaluated under both ImageNet OOD benchmark. Best performance highlighted in **bold**.

| Method | iNaturalist | | SUN | | Places | | Texture | | Avg | |
|---|---|---|---|---|---|---|---|---|---|---|
| | FPR95↓ | AUROC↑ | FPR95↓ | AUROC↑ | FPR95↓ | AUROC↑ | FPR95↓ | AUROC↑ | FPR95↓ | AUROC↑ |
| fDBD w/ResNet50 | 40.10 | 93.70 | 60.89 | 86.86 | 66.75 | 84.14 | 37.66 | 92.09 | 51.35 | 89.20 |
| fDBD w/ResNet50-supcon | 17.34 | 96.68 | 42.26 | 90.92 | 49.68 | 88.38 | 21.84 | 95.44 | 32.78 | 92.86 |
| fDBD w/ViT-B/16 | 12.97 | 97.71 | 51.09 | 89.67 | 56.51 | 87.32 | 45.62 | 89.48 | 41.55 | 91.05 |
| LAFO w/ResNet50 | 34.88 | 94.43 | 54.30 | 88.41 | 61.79 | 85.64 | 27.34 | 94.24 | 44.58 | 90.68 |
| LAFO w/ResNet50-supcon | 12.27 | 97.42 | 31.80 | 92.85 | 40.71 | 90.10 | 15.39 | 96.68 | 25.04 | 94.26 |
| LAFO w/ViT-B/16 | 11.81 | 97.85 | 48.98 | 90.06 | 54.60 | 87.75 | 44.31 | 89.85 | 39.92 | 91.38 |
| Ensemble fDBD | 4.58 | 98.93 | 42.81 | 93.97 | 53.49 | 91.92 | 23.33 | 96.34 | 31.05 | 95.29 |
| Ensemble LAFO | 2.77 | 99.29 | 30.21 | 95.39 | 42.52 | 93.39 | 14.63 | 97.59 | **22.53** | **96.41** |

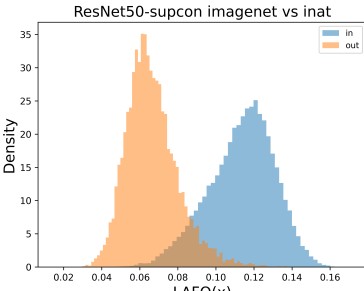 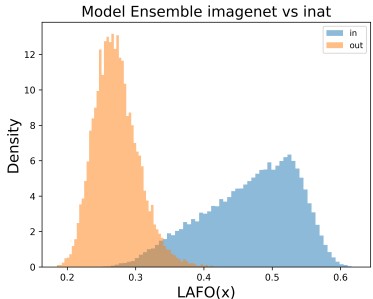

Figure 6: Comparison of the score histograms on Imagenet (ID) and inaturalist(Van Horn et al., 2018)(OOD) of the best individual model *(*left) with the model ensemble *(right)*. Model ensemble improves the ID and OOD separation.

### A.3 MODEL ENSEMBLE EXPERIMENT

Table 6 show the extended version for the model ensemble experiment presented on Table 3

### A.4 IMPLEMENTATION DETAILS

We used Pytorch (Paszke et al., 2019) to conduct our experiments. We obtain the checkpoints of pre-trained models ResNet18 with supervised contrastive loss and ResNet50 with supervised contrastive loss from Liu & Qin (2024)'s work for a fair comparison. In the experiment where we aggregate different models' confidences, ViT-B/16 (Dosovitskiy et al., 2020) checkpoint is retrieved from the publicly available repository https://github.com/lukemelas/PyTorch-Pretrained-ViT/tree/master. In the experiment where we merge LAFO with the activation shaping algorithms ASH (Djurisic et al., 2023), Scale (Xu et al.) and ReAct (Sun et al., 2021), we used the percentiles to set the thresholds 35, 90 and 80 respectively. All experiments are evaluated on a single Nvidia H100 GPU. Note that, due to our hyperparameter-free post-hoc score function, all experiments are deterministic given the pretrained model.

## B REBUTTAL EXPERIMENTS

Table 7: CIFAR10 Plain ResNet18 performances.

| Method | SVHN | | iSUN | | Places | | Texture | | Avg | |
|---|---|---|---|---|---|---|---|---|---|---|
| | FPR95↓ | AUROC↑ | FPR95↓ | AUROC↑ | FPR95↓ | AUROC↑ | FPR95↓ | AUROC↑ | FPR95↓ | AUROC↑ |
| KNN | 27.85 | 95.52 | 24.67 | 95.52 | 44.56 | 90.85 | 37.57 | 94.71 | 33.66 | 94.15 |
| fDBD | 22.58 | 96.07 | 23.96 | 95.85 | 46.59 | 90.40 | 31.24 | 94.48 | 31.09 | 94.20 |
| LAFO | 22.09 | 96.02 | 22.91 | 95.90 | 46.46 | 90.37 | 31.28 | 94.48 | **30.86** | **94.21** |

Table 8: ImageNet Plain ResNet50 performances.

| Method | iNaturalist | | SUN | | Places | | Texture | | Avg | |
|---|---|---|---|---|---|---|---|---|---|---|
| | FPR95↓ | AUROC↑ | FPR95↓ | AUROC↑ | FPR95↓ | AUROC↑ | FPR95↓ | AUROC↑ | FPR95↓ | AUROC↑ |
| KNN | 59.00 | 86.47 | 68.82 | 80.72 | 76.28 | 75.76 | 11.77 | 97.07 | 53.97 | 85.01 |
| fDBD | 40.24 | 93.67 | 60.60 | 86.97 | 66.40 | 84.27 | 37.50 | 92.12 | 51.19 | **89.26** |
| LAFO | 38.94 | 93.68 | 59.78 | 86.53 | 66.89 | 83.04 | 31.67 | 93.33 | **49.32** | 89.15 |

Table 9: ImageNet ViT performances.

| Method | iNaturalist | | SUN | | Places | | Texture | | Avg | |
|---|---|---|---|---|---|---|---|---|---|---|
| | FPR95↓ | AUROC↑ | FPR95↓ | AUROC↑ | FPR95↓ | AUROC↑ | FPR95↓ | AUROC↑ | FPR95↓ | AUROC↑ |
| KNN | 11.41 | 97.65 | 56.91 | 86.39 | 63.76 | 82.61 | 42.23 | 89.61 | 43.58 | 89.07 |
| fDBD | 12.86 | 97.72 | 50.86 | 89.74 | 56.28 | 87.44 | 45.74 | 89.41 | 41.44 | 91.08 |
| LAFO | 11.80 | 97.86 | 48.81 | 90.14 | 54.32 | 87.88 | 44.56 | 89.75 | **39.87** | **91.41** |

Table 10: LAFO vs ReAct under ImageNet OOD benchmark.

| Method | iNaturalist | | SUN | | Places | | Texture | | Avg | |
|---|---|---|---|---|---|---|---|---|---|---|
| | FPR95↓ | AUROC↑ | FPR95↓ | AUROC↑ | FPR95↓ | AUROC↑ | FPR95↓ | AUROC↑ | FPR95↓ | AUROC↑ |
| ReAct | 20.38 | 96.22 | 24.20 | 94.20 | 33.85 | 91.58 | 47.30 | 89.80 | 31.43 | 92.95 |
| LAFO | 12.27 | 97.42 | 31.80 | 92.85 | 40.71 | 90.10 | 15.39 | 96.68 | 25.04 | 94.26 |
| LAFO w/ReAct | 11.13 | 97.79 | 22.34 | 94.95 | 33.33 | 91.81 | 14.65 | 96.60 | **20.36** | **96.29** |

Table 11: LAFO vs ReAct under CIFAR OOD benchmark.

| Method | SVHN | | iSUN | | Places | | Texture | | Avg | |
|---|---|---|---|---|---|---|---|---|---|---|
| | FPR95↓ | AUROC↑ | FPR95↓ | AUROC↑ | FPR95↓ | AUROC↑ | FPR95↓ | AUROC↑ | FPR95↓ | AUROC↑ |
| ReAct | 6.15 | 98.75 | 10.31 | 98.09 | 21.68 | 95.47 | 10.18 | 98.12 | 12.08 | 97.61 |
| LAFO | 3.53 | 99.16 | 8.36 | 98.28 | 23.40 | 94.88 | 8.58 | 98.34 | 10.97 | 97.67 |
| LAFO w/ReAct | 3.35 | 99.18 | 8.11 | 98.29 | 20.84 | 95.25 | 7.87 | 98.45 | **10.04** | **97.79** |

Table 12: CIFAR10 centering with different statistics.

| Method | SVHN | | iSUN | | Places | | Texture | | Avg | |
|---|---|---|---|---|---|---|---|---|---|---|
| | FPR95↓ | AUROC↑ | FPR95↓ | AUROC↑ | FPR95↓ | AUROC↑ | FPR95↓ | AUROC↑ | FPR95↓ | AUROC↑ |
| Class 0 mean | 4.77 | 98.96 | 8.06 | 98.27 | 25.20 | 94.62 | 10.11 | 98.19 | 12.03 | 97.51 |
| Class 1 mean | 6.12 | 98.77 | 8.86 | 98.24 | 24.94 | 94.96 | 13.16 | 97.68 | 13.27 | 97.41 |
| Class 2 mean | 5.42 | 98.84 | 7.90 | 98.36 | 22.19 | 95.58 | 11.42 | 97.98 | 11.73 | 97.69 |
| Class 3 mean | 5.94 | 98.76 | 7.99 | 98.29 | 22.80 | 95.43 | 11.35 | 97.66 | 12.02 | 97.54 |
| Class 4 mean | 5.44 | 98.85 | 8.87 | 98.22 | 22.68 | 95.47 | 11.26 | 97.96 | 12.06 | 97.63 |
| Class 5 mean | 6.22 | 98.64 | 7.38 | 98.45 | 23.11 | 95.48 | 11.84 | 97.82 | 12.14 | 97.60 |
| Class 6 mean | 5.74 | 98.82 | 8.50 | 98.26 | 97.67 | 20.76 | 12.11 | 95.73 | 11.78 | 97.62 |
| Class 7 mean | 5.93 | 98.78 | 8.29 | 98.30 | 24.55 | 95.13 | 12.57 | 97.82 | 12.84 | 97.51 |
| Class 8 mean | 5.81 | 98.81 | 10.03 | 97.95 | 26.79 | 94.18 | 10.41 | 98.11 | 13.26 | 97.26 |
| Class 9 mean | 6.11 | 98.78 | 9.00 | 98.19 | 24.89 | 94.62 | 11.35 | 97.95 | 12.84 | 97.25 |
| Sum aggregation | 5.68 | 98.85 | 8.27 | 98.33 | 23.70 | 95.34 | 11.33 | 97.98 | 12.25 | 97.62 |
| Elementwise max | 6.28 | 98.73 | 8.28 | 98.30 | 13.79 | 97.57 | 24.35 | 95.21 | 13.18 | 97.45 |
| Elementwise min | 3.60 | 99.14 | 14.82 | 97.10 | 9.38 | 97.99 | 27.62 | 92.97 | 13.85 | 96.80 |
| Elementwise median | 2.33 | 99.34 | 10.02 | 97.90 | 7.73 | 98.29 | 23.99 | 93.84 | 11.02 | 97.34 |
| Sum aggregation | 5.78 | 98.65 | 20.31 | 95.64 | 10.35 | 97.80 | 30.42 | 91.60 | 16.72 | 95.92 |
| LAFO | 3.53 | 99.16 | 8.36 | 98.28 | 23.40 | 94.88 | 8.58 | 98.34 | **10.97** | **97.67** |

Table 13: ImageNet centering with different statistics.

| Method | iNaturalist | | SUN | | Places | | Texture | | Avg | |
|---|---|---|---|---|---|---|---|---|---|---|
| | FPR95↓ | AUROC↑ | FPR95↓ | AUROC↑ | FPR95↓ | AUROC↑ | FPR95↓ | AUROC↑ | FPR95↓ | AUROC↑ |
| Class 1 mean | 16.01 | 96.92 | 31.63 | 92.52 | 39.86 | 90.67 | 25.39 | 93.33 | 28.22 | 93.36 |
| Class 250 mean | 11.48 | 97.65 | 31.20 | 92.69 | 39.53 | 90.77 | 20.16 | 94.87 | 25.59 | 93.99 |
| Class 500 mean | 14.87 | 97.08 | 38.97 | 90.52 | 45.80 | 88.90 | 26.29 | 93.04 | 31.48 | 92.28 |
| Class 750 mean | 11.57 | 97.59 | 34.23 | 92.13 | 42.60 | 90.15 | 19.75 | 95.18 | 27.04 | 93.76 |
| Class 1000 mean | 11.36 | 97.63 | 30.20 | 93.01 | 38.12 | 91.08 | 19.31 | 95.31 | **24.75** | **94.26** |
| Sum aggregation | 12.40 | 97.48 | 32.47 | 92.36 | 40.42 | 90.53 | 21.38 | 94.52 | 26.67 | 93.72 |
| Elementwise Max | 17.10 | 96.76 | 34.22 | 91.73 | 41.88 | 90.14 | 35.09 | 89.84 | 32.07 | 92.12 |
| Elementwise Min | 29.16 | 94.51 | 60.70 | 85.81 | 65.01 | 83.18 | 22.84 | 95.07 | 44.43 | 89.64 |
| Elementwise Median | 20.04 | 95.83 | 46.66 | 89.15 | 54.42 | 85.61 | 15.04 | 96.81 | 34.04 | 91.85 |
| Sum aggregation | 21.09 | 95.89 | 49.47 | 88.97 | 56.50 | 86.12 | 17.62 | 95.98 | 36.17 | 91.74 |
| LAFO | 12.27 | 97.42 | 31.80 | 92.85 | 40.71 | 90.10 | 15.39 | 96.68 | 25.04 | **94.26** |

Table 14: Resource Constrained Setting: ImageNet MobileNet_v2 performances.

| Method | iNaturalist | | SUN | | Places | | Texture | | Avg | |
|---|---|---|---|---|---|---|---|---|---|---|
| | FPR95↓ | AUROC↑ | FPR95↓ | AUROC↑ | FPR95↓ | AUROC↑ | FPR95↓ | AUROC↑ | FPR95↓ | AUROC↑ |
| MSP | 59.84 | 86.71 | 74.15 | 78.87 | 76.84 | 78.14 | 70.98 | 78.95 | 70.45 | 80.67 |
| Energy | 55.35 | 90.33 | 59.36 | 86.24 | 66.28 | 83.21 | 54.54 | 86.58 | 58.88 | 86.59 |
| KNN | 85.92 | 72.67 | 90.51 | 65.39 | 93.21 | 60.08 | 14.04 | 96.98 | 70.92 | 73.78 |
| fDBD | 53.72 | 90.89 | 68.22 | 82.84 | 73.20 | 80.09 | 37.82 | 91.85 | 58.24 | 86.42 |
| LAFO | 46.59 | 91.86 | 61.21 | 85.01 | 67.81 | 82.08 | 27.07 | 94.04 | **50.68** | **88.25** |

Table 15: ImageNet CLIP-ViT-H14 performances. (LP: Linear Probe, ZS: Zero Shot)

| Method | iNaturalist | | SUN | | Places | | Texture | | Avg | |
|---|---|---|---|---|---|---|---|---|---|---|
| | FPR95↓ | AUROC↑ | FPR95↓ | AUROC↑ | FPR95↓ | AUROC↑ | FPR95↓ | AUROC↑ | FPR95↓ | AUROC↑ |
| MSP (LP) | 15.74 | 96.64 | 46.00 | 88.68 | 48.73 | 87.40 | 40.87 | 87.98 | 37.83 | 90.18 |
| Energy (LP) | 7.26 | 97.94 | 34.62 | 92.13 | 41.32 | 90.05 | 37.02 | 90.98 | 30.06 | 92.77 |
| KNN (ZS) | 80.20 | 87.86 | 86.68 | 84.63 | 73.51 | 86.07 | 70.27 | 84.60 | 77.66 | 85.79 |
| fDBD (ZS) | 9.31 | 98.11 | 22.32 | 94.78 | 29.15 | 93.20 | 40.12 | 90.25 | 25.23 | 94.08 |
| fDBD (LP) | 5.62 | 98.48 | 32.18 | 93.89 | 35.74 | 92.54 | 27.13 | 93.71 | 25.17 | 94.66 |
| LAFO (ZS) | 14.12 | 97.41 | 22.97 | 94.97 | 28.01 | 93.41 | 38.28 | 90.73 | 25.85 | 94.13 |
| LAFO (LP) | 6.66 | 98.16 | 30.35 | 94.43 | 33.79 | 93.20 | 24.95 | 94.34 | **23.94** | **95.03** |

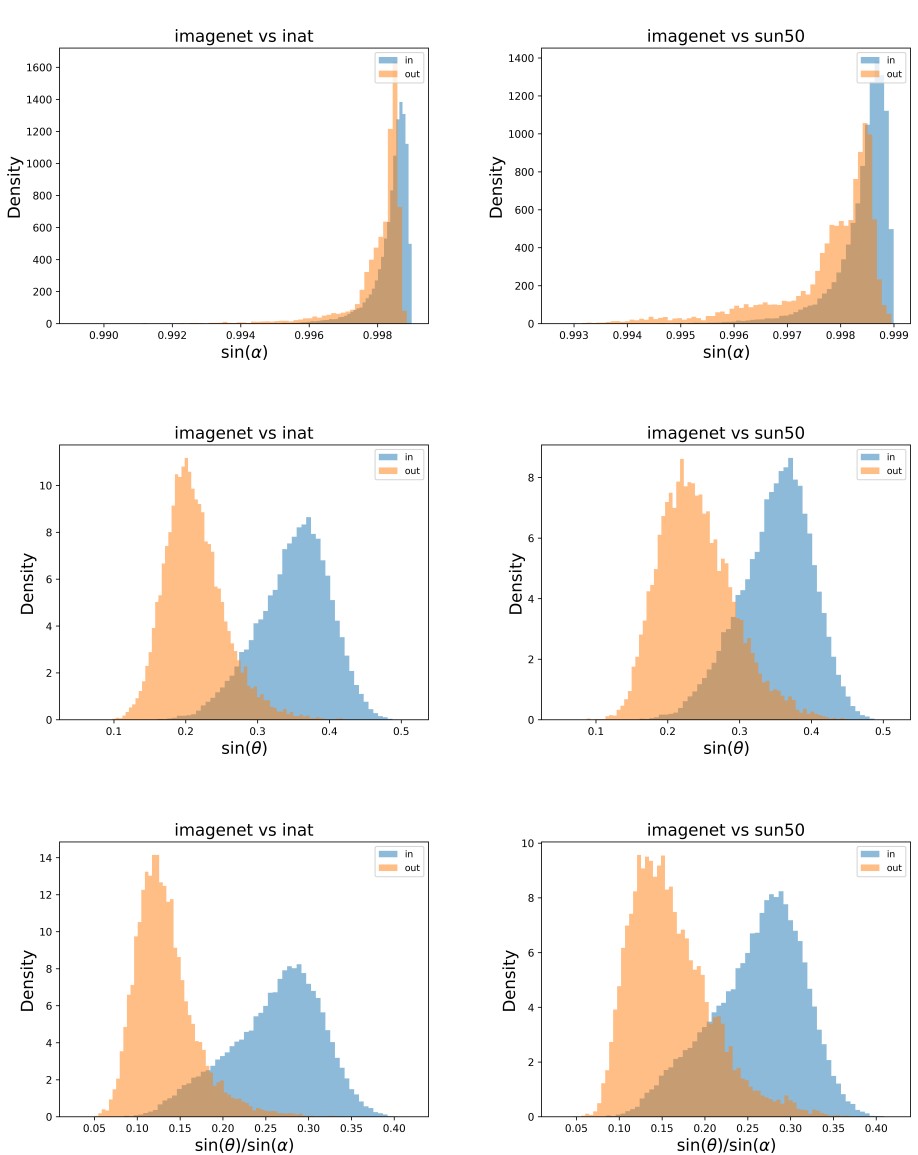

Figure 7: We demonstrate the ID/OOD separability of $\sin(\alpha)$, $\sin(\theta)$ and $\frac{\sin(\theta)}{\sin(\alpha)}$. Columns show the performances on iNaturalist and SUN datasets respectively. It can be seen that the ID/OOD class separability is the best when $\sin(\theta)$ is used: considering $\sin(\alpha)$ impedes the performance as confirmed quantitatively in terms of FPR95 and AUROC metrics in Table 2.

