# OpenReview forum: "Look Around and Find Out: OOD Detection with Relative Angles"
_ICLR.cc/2025/Conference — Submitted to ICLR 2025_

### Official Review · Reviewer_Vyey · 2024-10-31

**Soundness:** 2
**Presentation:** 3
**Contribution:** 2
**Rating:** 6
**Confidence:** 4

**Summary:**

This work proposed a novel angle-based metric for OOD detection that is computed relative to the in-distribution structure. They demonstrate that the angles between feature representations and decision boundaries, viewed from the mean of in-distribution features, serve as an effective discriminative factor between ID and OOD data. Experiments on CIFAR10 and ImageNet shows SOTA performance compared to other detection methods.

**Strengths:**

The performance is good and the analysis is easy to understand.
The metric is scale-invariant, allowing ensemble for better performance.
The experiment is comprehensive.

**Weaknesses:**

1. From Figure 1, the angle \alpha is helpful for better distinguishing ID and OOD data. And there lacks a comparison between the sine of \alpha, the sine of \theta, and the division of them. I think only the sine of \alpha in Figure 2 is not convincing to demonstrate that the angle \alpha is not very informative for ID and OOD separation.

2. For experiments, I think the author should report detection results on a vanilla trained model, which is a more common and practical setting for post-hoc detection methods. The current results are all based on supervised contrastive training models.

3. For experiments, it should compare with ReAct method on ImageNet OOD benchmark (in Table 2), since my empirical experience tells that ReAct always shows remarkable performance on ImageNet dataset.

**Questions:**

no more question

---

> ### Author Response · Authors · 2024-11-20
> **Response to Reviewer Vyey**
>
> **Histograms for LAFO and fDBD:**  Note that $\sin(\\theta) / \sin(\alpha)$ and $\sin(\theta)$ correspond to fDBD \[8\] and LAFO scores respectively. We provide the additional histograms on Figure 7 by picking the datasets where the performance difference is visible on histograms. It can be seen that the division by $\sin(\alpha)$ impedes the ID/OOD separability.
>
> **Plain architectures:** On tables 7 and 8 we report the performances with vanilla trained ResNets \[1\]. Results show that LAFO consistently outperforms these baselines on vanilla trained architectures as well.
>
> **Including ReAct, and additional comparisons:** We include ReAct \[15\] as a baseline on our main tables 1 and 2\. Moreover, we report the detailed performances of our method, ReAct and the combination on Tables 10 and 11\. LAFO demonstrates better performance than ReAct on both the Imagenet and CIFAR benchmarks. The best performance is achieved by incorporating ReAct in LAFO.

---

> > ### Author Response · Authors · 2024-11-25
> >
> > We thank the Reviewer once again for their valuable feedback.
> >
> > We hope our rebuttal has addressed their concerns, particularly in light of the additional experiments conducted during the rebuttal period.
> >
> > We remain available to address any further questions or concerns during the discussion period.

---

> > > ### Comment · Reviewer_MwRp · 2024-11-26
> > > **Could be further improved.**
> > >
> > > Thanks for the authors' response, some of my concerns are addressed. Yet, I believe the paper should be further improved following reviewers' suggestions, thus I decide to keep my score.

---

> > > ### Comment · Reviewer_Vyey · 2024-11-26
> > >
> > > Thanks for your detailed response. I decide to raise my score to 6.

---

> ### Author Response · Authors · 2024-11-26
> **Suggestions on improvement**
>
> Thank you for your response. We are happy to address any specific suggestions for improving the paper.
>
> Following the suggestions of other reviewers, we conducted an extensive set of experiments, which demonstrate that LAFO consistently outperforms other methods across all requested settings. Please refer to our shared response for details and the "Rebuttal Experiments" section in the Appendix for the tables.

---

### Official Review · Reviewer_MwRp · 2024-11-01

**Soundness:** 2
**Presentation:** 3
**Contribution:** 2
**Rating:** 5
**Confidence:** 5

**Summary:**

This paper presents Look Around and Find Out (LAFO), a novel approach for out-of-distribution (OOD) detection using angle-based metrics. By calculating angles between feature representations and decision boundaries in relation to the mean of in-distribution (ID) features, LAFO improves OOD detection performance by leveraging the geometric relationships within feature space. The proposed method demonstrates robust performance across multiple benchmarks (CIFAR-10, ImageNet), significantly reducing false positive rates (FPR95) compared to state-of-the-art methods. Additionally, LAFO is hyperparameter-free, scale-invariant, and compatible with ensemble models, which enhances its practical utility.

**Strengths:**

- The angle-based approach relative to ID mean is novel in differentiating ID and OOD samples.
- LAFO’s lack of hyperparameters simplifies its use in practical scenarios and avoids overfitting issues associated with tuning.
- The model achieves impressive results on CIFAR-10 and ImageNet, showcasing its scalability from smaller to larger datasets.
- LAFO can be combined with other activation shaping methods, demonstrating flexibility in enhancing model confidence scores.

**Weaknesses:**

- While effective, using only the ID mean for centering may limit adaptability across highly variable datasets. Incorporating other statistics could improve robustness.
- The experiments focus on ResNet architectures. Additional comparisons with transformer-based or CLIP-based architectures could provide more insights.
- The paper does not fully explore scenarios where LAFO may struggle, such as in cases with minimal separability between ID and OOD distributions.
- Although LAFO is efficient, the paper could address its performance in real-time or resource-constrained settings to provide a more comprehensive view.

**Questions:**

The effectiveness of LAFO in scenarios with severe ID class overlapping?

---

> ### Author Response · Authors · 2024-11-20
> **Response to Reviewer MwRp**
>
> **Resource constrained setting Mobilenet v2:**
> To evaluate LAFO’s performance in resource-constrained settings, we applied our method on the MobileNet v2 \[4\] – a model designed for efficient inference with minimal resources. Table 14 shows the performance of LAFO alongside baselines. LAFO performs best on average according to FPR and AUROC metrics, demonstrating the adaptability to resource constrained settings.
>
> **Clarification on problem setting about ID/OOD separability:** Following the previous work \[8, 15, 16, 17, 18\] and the problem setting described by \[19\]
>
> >“... in the OOD detection, the test samples come from a distribution whose semantics are shifted from ID, i.e.,P(Y) $\neq$ P (Y').”
>
> We evaluated our score mechanism on the common benchmarks where there are not common classes between ID and OOD. As suggested by these, it is essential to ensure mutually exclusive classes between ID and OOD to be able to evaluate the performance of the model objectively.
>
> **ID Class Overlap:** We believe ImageNet \[10\] is a dataset with significant class overlap, as some classes represent different species of the same animal, such as “Siberian Husky” vs. “Malamute” or “Grasshopper” vs. “Cricket.” Despite this challenge, LAFO demonstrates the best performance on the ImageNet benchmark by a significant margin, highlighting its robustness even in scenarios with severe class overlap.

---

> > ### Author Response · Authors · 2024-11-25
> >
> > We thank the Reviewer once again for their valuable feedback.
> >
> > We hope our rebuttal has addressed their concerns, particularly in light of the additional experiments conducted during the rebuttal period.
> >
> > We remain available to address any further questions or concerns during the discussion period.

---

> > > ### Author Response · Authors · 2024-11-26
> > > **Clarification on Reviewer Response**
> > >
> > > We are copying our response to the reviewer MwRp on this tread because they had answered (and we replied) under another reviewer's comment.
> > >
> > > Thank you for your response. We are happy to address any specific suggestions for improving the paper.
> > >
> > > Following the suggestions of other reviewers, we conducted an extensive set of experiments, which demonstrate that LAFO consistently outperforms other methods across all requested settings. Please refer to our shared response for details and the "Rebuttal Experiments" section in the Appendix for the tables.

---

### Official Review · Reviewer_Qz8h · 2024-11-04

**Soundness:** 3
**Presentation:** 2
**Contribution:** 3
**Rating:** 6
**Confidence:** 3

**Summary:**

This paper presents a novel method for out-of-distribution (OOD) detection based on feature representations in neural networks. The proposed approach, LAFO (Look Around and Find Out), introduces an angle-based metric that measures the angle between feature representations and decision boundaries relative to the mean in-distribution (ID) feature. This approach leverages the relationship between feature vectors and decision boundaries to differentiate between ID and OOD samples effectively.

**Strengths:**

1. The idea sounds novel. The paper introduces a novel angle-based metric for OOD detection, which measures the angle between feature representations and decision boundaries relative to the mean of in-distribution (ID) data.

2. This paper conducts extensive experiments to validate the proposed approach, including the standard benchmarks, and demonstrates its flexibility by incorporating it into ensemble methods and combining it with activation shaping algorithms.

3. This paper also explained the connection between LAFO and the similar approach fDBD.

**Weaknesses:**

1. The exploration of ID statistics beyond the mean is limited. As OOD detection can benefit from a richer representation of ID statistics, does the "ID mean" refer to the mean across all classes? How about class-specific means or other statistical summaries? If the author includes these experiments or analyses, the paper will be strengthened.

2. The experiments do not sufficiently address why and how LAFO enhances ensemble performance compared to other methods. It would be beneficial to see a more detailed analysis of how the angle-based scores behave in various ensemble settings, such as different architectures or training losses, to better understand when and why LAFO performs optimally.

**Questions:**

1. Why is the CSI baseline used for CIFAR10 OOD benchmark, but not used for imageNet benchmark?

2. Did you consider the application of LAFO on multi-modal foundation models, such as CLIP?

3. The feature is evaluated through the composed function $f_1\circ...\circ f_{L-1} \circ g $, can you explain the reason why using this way or show some references for this? Did you consider other ways to represent features?

---

> ### Author Response · Authors · 2024-11-20
> **Response to Reviewer Qz8h**
>
> **Model ensembling:**  Figure 3 demonstrates the performance of LAFO and the fDBD \[8\] baseline across different architectures (ResNet50 \[1\], ResNet50 with supervised contrastive loss \[21\], and ViT \[2\]). LAFO consistently outperforms fDBD, both on individual architectures, and in ensemble settings showing its effectiveness.
>
> **Suitability for ensembling:** The key reason for LAFO is suitable for ensembles lies in its angle-based representation. Recent work \[22\] has shown that angle-based representations effectively capture relationships between feature spaces of models with different architectures and pre-training strategies. This makes LAFO a natural choice for model ensembling in OOD detection, compared to methods utilizing less compatible representations.
>
> **CSI Imagenet performance:** The CSI \[14\] baseline was not used for the ImageNet \[10\] OOD benchmark due to its significant computational demands. In the original paper where CSI was proposed, performance on the ImageNet OOD benchmark was not reported. Additionally, in the related work, kNN \[7\], they report that:
>
> >"The training procedure of CSI is computationally prohibitive on ImageNet, which takes three months on 8 Nvidia 2080Tis."
>
> Given these substantial resource requirements, we chose not to include CSI in our analysis.
>
> **Clarification on feature representation notation:** With the composite function we simply refer to the features extracted at the penultimate layer of the deep classifier, which is the standard way to select representations \[7, 8, 13, 20\]. We will make this clearer in the manuscript, thank you for pointing this out

---

> > ### Author Response · Authors · 2024-11-25
> >
> > We thank the Reviewer once again for their valuable feedback.
> >
> > We hope our rebuttal has addressed their concerns, particularly in light of the additional experiments conducted during the rebuttal period.
> >
> > We remain available to address any further questions or concerns during the discussion period.

---

> ### Comment · Reviewer_Qz8h · 2024-11-26
>
> Thanks for the explanation in the response! Most concerns were clarified. Though I have two more questions:
>
> 1) Different statistics: I appreciate the authors' efforts in conducting more experiments. Could you please provide some insights or explanations about the current results? My impression is that many experiments show the effectiveness of the proposed approach. However, we might not have a better understanding or intuition why the method works. High-level explanation is accepted.
>
> 2) The experiments of CLIP: Did you consider multi-modal (vision-language) on zero-shot OOD detection [1], or just used the backbone of vision encoder in your experiments? If just used the vision encoder, is that possible to try the setting of zero-shot OOD detection considering multiple modalities [1] based on ImageNet (or ImageNet-200)?
>
> [1] Yifei Ming, Ziyang Cai, Jiuxiang Gu, Yiyou Sun, Wei Li, and Yixuan Li.Delving into out-of-distribution detection with vision-language representations.Advances in Neural Information Processing Systems, 2022.

---

> > ### Author Response · Authors · 2024-11-27
> >
> > We thank the Reviewer for their constructive feedback!
> >
> > ### **Different Statistics:**
> >
> > The angle looking at the line segment between the feature representation of the sample and its projection onto the decision boundary is the key metric we consider. Using angles instead of absolute distances allows us to focus on how much a sample's representation diverges directionally from ID features, independent of magnitude. Moreover, it comes with additional desired properties like scale-invariance (which allows us to compare and aggregate the decisions of different models).
> >
> > **What happens if we just look from the origin?**
> >
> > After opting for an angular score, it is important that from which reference frame we are looking at the line segment of our interest. A natural choice could be the origin. However, angles measured from the origin would be sensitive to constant shifts in the representations, even though the geometric relationships and representation quality remain unchanged.
> >
> > Instead, one can move every representation to the same orthant, then measure the angle from the origin. Note that this angle ranking would be identical to the absolute distance. Nevertheless, this choice would result in angle measurements that are larger whenever the representation is closer to the origin after shifting. Therefore, it would give different weights to different classes depending on their position.
> >
> > **Why do we choose the mean?**
> >
> > We address this issue by selecting the mean of the ID features as the reference point. This approach avoids overemphasizing any particular class and provides an unbiased estimate of the central tendency of the ID feature distribution. Specifically, we first compute the mean for each class and then take the average across these class-level means. This ensures that imbalanced datasets do not disproportionately influence the reference point.
> >
> >
> >
> > ### **CLIP Experiments**
> >
> > We primarily use the vision encoder of CLIP with a linear probe trained for ImageNet classification, which aligns with our need for well-defined decision boundaries. In contrast, with cosine similarity-based decisions, finding decision boundaries would require a slightly different formulation. In response to the request, we applied the following derivation to enable the use of LAFO for zero-shot OOD detection with CLIP.
> >
> >
> > **Zero-shot OOD Detection with CLIP**
> >
> > To obtain the projection onto the decision boundary, we study the normalized representations for simplicity.
> >
> > Let $z_{c_1}$ and $z_{c_2}$ be the text representations of two different classes. We are interested in obtaining the projection of feature representation z, onto the decision boundary between these two classes.
> >
> > Decisions for classes $c_1$ and $c_2$ are equally strong for some representation $t$, if
> >
> > $$ \langle t, z_{c_1} \rangle = \langle t, z_{c_2} \rangle$$
> > $$ \langle t,  (z_{c_1} - z_{c_2}) \rangle = 0$$
> >
> > Define $u = \frac{(z_{c_1} - z_{c_2})}{||(z_{c_1} - z_{c_2})||}$. Then, to obtain the representation on the decision boundary, we can project $z$ onto the orthogonal subspace by:
> >
> > $$ z_{db} = z - \langle z, u \rangle \cdot u$$
> >
> > From there, we can normalize $z_{db}$ and calculate our score, as described in the paper.
> >
> > **Results:** We report the zero-shot performances of LAFO and fDBD by extending Table 15 in the Rebuttal Experiments section. It is indeed possible to apply CLIP zero-shot. However, the euclidean distance on the constrained unit-norm representation space does not carry the interpretations we initially begin with. Nevertheless, LAFO achieves 25.85 FPR95 and 94.13 AUROC in this setting, showing its flexibility. From the table, it can be seen that the performances of fDBD and LAFO are very similar, while on average falling behind in FPR95, LAFO is better on AUROC, demonstrating a competitive performance on this setting.

---

> > > ### Comment · Reviewer_Qz8h · 2024-11-27
> > >
> > > Thanks for the authors' efforts in the response.  I decide to raise my score to 6. The authors might need to ensure the equations are shown appropriately before posting. (though I cannot understand 100%)

---

### Official Review · Reviewer_Dg6s · 2024-11-04

**Soundness:** 3
**Presentation:** 3
**Contribution:** 3
**Rating:** 5
**Confidence:** 4

**Summary:**

The paper proposes to calculate the angle between the feature representation and the decision boundary, viewing from the mean of ID representations, to compute a score for identifying OOD examples. The method is evaluated on two popular benchmarks: CIFAR100 and Imagenet for OOD detection.

**Strengths:**

1. The paper is well-written and easy to follow.

2. Relying on the angle between feature representations and the decision boundary seems to be novel.

3. The geometric interpretation of the presented method is convincing.

4. The presented method can easily be integrated into existing frameworks.

**Weaknesses:**

1. I am somewhat skeptical about the performance gain. Although the paper claim performance gains across both benchmarks, the improvement is marginal for CIFAR100 (0.8% FPR95) and only evident on average. Looking at Table 1 and Table 2, the method lags behind other methods on an individual basis. It’s important to discuss why the method does not generalize well on an individual basis.

2. The method is only compared on ResNet architectures. How does it perform on other recent architectures, such as Vision Transformers? Given that the method relies heavily on feature and decision boundaries, validating it on diverse architectures is essential to confirm its architecture-agnostic and plug-in characteristics.

Minor Fixes: Please review the references. Some include only the publication year without the publication venue.

**Questions:**

Mostly, my concerns are on performance gain and the experiments on different architectures. If authors can convincingly address my concerns, I am willing to change my rating.

---

> ### Author Response · Authors · 2024-11-20
> **Response to Reviewer Dg6s**
>
> **Performance on individual datasets:**
>
> - On the large-scale ImageNet \[10\] OOD benchmark (Table 2), LAFO beats every method on the table significantly on every individual dataset (gaining up to \~9% compared to the second best), making our method state-of-the-art consistently on individual datasets and on average .
> - On the small-scale CIFAR \[9\] benchmark (Table 1), LAFO shows marginally weaker performance on some individual datasets but remains the best on average. Reasons include:
>   - The benchmark is near saturation (e.g., \~97.5 average AUROC across top methods and \~99.5 on SUN \[11\] and SVHN \[12\]), making further gains negligible.
>   - Small improvements on specific datasets do not guarantee better overall OOD detection. For example, while LAFO lags slightly (\~2%) behind SSD+ \[13\] on SVHN, it outperforms significantly (up to \~25%) on most other datasets.
>   - For methods that perform best on individual datasets (e.g., SSD+\[13\] on SVHN), there is typically at least one other dataset where their performance is poor, highlighting weak generalization across multiple distribution shifts. In contrast, LAFO demonstrates strong performance across all datasets, effectively generalizing to multiple distribution shifts. This makes it a more suitable choice for OOD detection in real-world scenarios, where multiple unknown distribution shifts are likely to occur.

---

> > ### Author Response · Authors · 2024-11-25
> >
> > We thank the Reviewer once again for their valuable feedback.
> >
> > We hope our rebuttal has addressed their concerns, particularly in light of the additional experiments conducted during the rebuttal period.
> >
> > We remain available to address any further questions or concerns during the discussion period.

---

> > > ### Author Response · Authors · 2024-11-27
> > >
> > > As the deadline for updating the manuscript is approaching, we kindly ask the reviewer to let us know if our new experiments are satisfactory (also see general response with ViT, CLIP and MobileNet results).

---

> > > > ### Author Response · Authors · 2024-12-02
> > > >
> > > > As the discussion period deadline approaches, we kindly ask the reviewer to let us know if any further clarifications are needed. We are happy to provide additional details or address any concerns.

---

### Author Response · Authors · 2024-11-20
**General response to all reviewers and the AC**

We would like to thank all reviewers for their comments, suggestions, and valuable feedback. We very much appreciate that they consider our submission “ *novel* ” (all reviewers), with “*comprehensive experiments*” (Qz8h, Vyey) “, “*impressive and scalable results*” (MwRp) and “*flexible*”  (Qz8h, Dg6s, MwRp).

Below we address common questions and concerns among reviewers. For individual questions and concerns, we are going to reply to each reviewer separately. We remain available for any further questions or concerns during the rebuttal period.

We have included a section titled 'Rebuttal Experiments' in the Appendix and are working on integrating it into the main paper.

**Performance on different architectures (Reviewer Dg6s, Reviewer MwRp):** In the model ensemble experiment (Figure 3), we demonstrated that LAFO generalizes well to multiple architectures, including ResNet and Vision Transformers (ViT). In this rebuttal, we provide additional detailed results for more architectures:

- ResNet, without supervised contrastive loss (Tables 7 and 8\) \[1\]
- ViT (Table 9\) without ensembling \[2\]
- CLIP (Table 15\) \[3\]
- MobileNet v2 (Table 14\) \[4\]

LAFO consistently achieves the best performance compared to baseline methods (e.g., Energy \[5\], MSP \[6\], kNN \[7\], fDBD \[8\]) across all these architectures, demonstrating its architecture-agnostic and plug-in characteristics.

**Performance analysis on different ID statistics (Reviewer Qz8h, Reviewer MwRp):** We thank the reviewers for highlighting this valuable direction. In the ablation in Table 4 in the paper we considered using the predicted and the target class mean statistics to show the performance of dynamic reference view. Following the suggestions, we included additional ablations in this rebuttal analyzing alternative sources of ID statistics beyond the ID mean, including:

- Individual class means
- Sum aggregation over the scores of individual class means
- Feature-wise minimum, maximum, and median of ID features
- Sum aggregation of minimum, maximum, and median


Detailed results (Tables 12 and 13\)  show LAFO with ID mean statistics consistently performs better than or on par with these alternative statistics on CIFAR \[9\] and ImageNet \[10\] OOD benchmarks.

**Future direction:** Exploring further alternatives, such as n-th-order moments of the ID statistics, remains a promising direction, as incorporating them in the formulation of the score is not trivial.

**Experiment with CLIP (Reviewer Qz8h, Reviewer MwRp):** We applied LAFO to the CLIP \[3\] architecture, specifically CLIP-ViT-H14.

To access decision boundaries, we trained a linear probe on CLIP features. This aligns with the premise of OOD detection, as most score functions (except kNN \[7\]) inherently rely on logits or a classification head.

Table 15 demonstrates that LAFO significantly outperforms strong baselines such as MSP \[6\], Energy \[5\], kNN \[7\], and fDBD \[8\], achieving the best performance on both FPR95 and AUROC.

To conclude, we would like to remark that for this experiment, we cannot guarantee that the classes in our OOD benchmark datasets are completely distinct from those in CLIP’s training data and similar concepts may be present in the open vocabulary textual descriptions used to learn the features.

---

> ### Author Response · Authors · 2024-11-20
> **Reference List**
>
> \[1\] Kaiming He, Xiangyu Zhang, Shaoqing Ren, and Jian Sun. Deep residual learning for image recognition. In Proceedings of the IEEE conference on computer vision and pattern recognition, pp. 770–778, 2016\.
> \[2\] Alexey Dosovitskiy, Lucas Beyer, Alexander Kolesnikov, Dirk Weissenborn, Xiaohua Zhai, Thomas Unterthiner, Mostafa Dehghani, Matthias Minderer, Georg Heigold, Sylvain Gelly, et al. An image is worth 16x16 words: Transformers for image recognition at scale. In International Conference on Learning Representations, 2020
> \[3\] Alec Radford, Jong Wook Kim, Chris Hallacy, Aditya Ramesh, Gabriel Goh, Sandhini Agarwal, Girish Sastry, Amanda Askell, Pamela Mishkin, Jack Clark, et al. Learning transferable visual models from natural language supervision. ICML, 2021\.
> \[4\] Mark Sandler, Andrew Howard, Menglong Zhu, Andrey Zhmoginov, and Liang-Chieh Chen. Mobilenetv2: Inverted residuals and linear bottlenecks. In Proceedings of the IEEE Conference on Computer Vision and Pattern Recognition (CVPR), June 2018\.
> \[5\] Weitang Liu, Xiaoyun Wang, John Owens, and Yixuan Li. Energy-based out-of-distribution detection. Advances in neural information processing systems, 33:21464–21475, 2020\.
> \[6\] Dan Hendrycks and Kevin Gimpel. A baseline for detecting misclassified and out-of-distribution examples in neural networks. In International Conference on Learning Representations, 2022\.
> \[7\] Sun, Y., Ming, Y., Zhu, X., & Li, Y. (2022, June). Out-of-distribution detection with deep nearest neighbors. In International Conference on Machine Learning (pp. 20827-20840). PMLR.
> \[8\] Liu, L., & Qin, Y. Fast Decision Boundary based Out-of-Distribution Detector. In Forty-first International Conference on Machine Learning.
> \[9\] Alex Krizhevsky, Geoffrey Hinton, et al. Learning multiple layers of features from tiny images. 2009
> \[10\] Jia Deng, Wei Dong, Richard Socher, Li-Jia Li, Kai Li, and Li Fei-Fei. Imagenet: A large-scale hierarchical image database. In 2009 IEEE conference on computer vision and pattern recognition, pp. 248–255. Ieee, 2009\.
> \[11\] Jianxiong Xiao, James Hays, Krista A Ehinger, Aude Oliva, and Antonio Torralba. Sun database: Large-scale scene recognition from abbey to zoo. In 2010 IEEE computer society conference on computer vision and pattern recognition, pp. 3485–3492. IEEE, 2010\.
> \[12\] Yuval Netzer, Tao Wang, Adam Coates, Alessandro Bissacco, Baolin Wu, Andrew Y Ng, et al. Reading digits in natural images with unsupervised feature learning. In NIPS workshop on deep learning and unsupervised feature learning, volume 2011, pp. 4\. Granada, 2011\.
> \[13\] Sehwag, V., Chiang, M., & Mittal, P. SSD: A Unified Framework for Self-Supervised Outlier Detection. In International Conference on Learning Representations.
> \[14\] Tack, J., Mo, S., Jeong, J., & Shin, J. (2020). Csi: Novelty detection via contrastive learning on distributionally shifted instances. Advances in neural information processing systems, 33, 11839-11852.
> \[15\] Sun, Y., Guo, C., & Li, Y. (2021). React: Out-of-distribution detection with rectified activations. Advances in Neural Information Processing Systems, 34, 144-157.
> \[16\] Djurisic, A., Bozanic, N., Ashok, A., & Liu, R. Extremely Simple Activation Shaping for Out-of-Distribution Detection. In The Eleventh International Conference on Learning Representations.
> \[17\] Wei, H., Xie, R., Cheng, H., Feng, L., An, B., & Li, Y. (2022, June). Mitigating neural network overconfidence with logit normalization. In International conference on machine learning (pp. 23631-23644). PMLR.
> \[18\] Sun, Y., & Li, Y. (2022, October). Dice: Leveraging sparsification for out-of-distribution detection. In European Conference on Computer Vision (pp. 691-708). Cham: Springer Nature Switzerland.
> \[19\] Yang, J., Zhou, K., Li, Y., & Liu, Z. (2024). Generalized out-of-distribution detection: A survey. International Journal of Computer Vision, 1-28.
> \[20\] Kimin Lee, Kibok Lee, Honglak Lee, and Jinwoo Shin. A simple unified framework for detecting out-of-distribution samples and adversarial attacks. Advances in neural information processing systems, 31, 2018b.
> \[21\] Prannay Khosla, Piotr Teterwak, Chen Wang, Aaron Sarna, Yonglong Tian, Phillip Isola, Aaron Maschinot, Ce Liu, and Dilip Krishnan. Supervised contrastive learning. Advances in neural information processing systems, 33:18661–18673, 2020\.
> \[22\] Luca Moschella, Valentino Maiorca, Marco Fumero, Antonio Norelli, Francesco Locatello, and Emanuele Rodola. Relative representations enable zero-shot latent space communication. In \` The Eleventh International Conference on Learning Representations.

---

### Meta-Review · Area_Chair_34SA · 2024-12-19

**Metareview:**

The LAFO (Look Around and Find Out) paper proposes an angle-based metric for out-of-distribution (OOD) detection. It calculates the angle between feature representations and their projection onto decision boundaries, relative to the mean of in-distribution features. Experiments on CIFAR-10 and ImageNet benchmarks demonstrate improved OOD detection performance compared to existing methods, including fDBD, showcasing LAFO's practicality and effectiveness.

However, while LAFO improves upon fDBD, the advancements could be seen as incremental rather than groundbreaking. Using angles instead of distances and mean-centering are relatively straightforward modifications, not fundamentally new paradigms. More importantly, this paper does not deeply investigate why angles work better than distances in OOD detection. The argument for using angles feels somewhat empirical and less grounded in theory, which weakens the perceived depth of the contribution.

**Additional Comments On Reviewer Discussion:**

The AC really appreciates the efforts made by the authors in the rebuttal, which managed to raise the score of a few reviewers. However, after reading all the reviews, the rebuttal, the paper and considering the most relevant prior works, it seems the advancements brought by this paper are relatively incremental.

---

### Decision · Program_Chairs · 2025-01-22

Reject